# Enabling liquid crystal elastomers with tunable actuation temperature

Yanjin Yao[1], Enjian He[1], Hongtu Xu[1], Yawen Liu[1], Zhijun Yang[1], Yen Wei ⓘ[1,2] & Yan Ji ⓘ[1] ✉

Liquid crystalline elastomers are regarded as a kind of desirable soft actuator material for soft robotics and other high-tech areas. The isotropization temperature ($T_i$) plays an important role as it determines the actuation temperature and other properties, which in turn has a great effect on their applications. In the past, the common physical methods (e.g. annealing) to tune $T_i$ is not applicable to tune the actuation temperature. The new $T_i$ obtained by annealing immediately goes back to the old one once it is heated to a temperature above $T_i$, while actuation needs a temperature higher than $T_i$. For a fully cross-linked LCE material, once it is synthesized, the actuation temperature is fixed. Accordingly, the actuation temperature can not be tuned unless the chemical structure is changed, which usually needs to start from the very beginning of the molecular design and material synthesis. Here, we found that different $T_i$ achieved by annealing can be preserved by reversible reactions of dynamic covalent bonds in covalently adaptable LC networks including LC vitrimers. Thus, a variety of soft actuators with different actuation temperatures can be obtained from the same fully cross-linked LCE material. As the tuning of $T_i$ is also reversible, the same actuator can be adjusted for applications with different actuation temperature requirements. Such tuning will also expand the application of LCEs.

Liquid crystal elastomers (LCEs) are of rapidly growing interest in recent years as diverse applications[1–5] have been proposed in various areas such as robotics[6–8], micromachines[9–11], artificial muscles[12], and tissue engineering[13], to name a few. It looks like there is no limit to their potential so long as researchers have enough imagination. One reason for the broad application prospects is that LCEs are capable of dramatic reversible deformation upon thermal, optical, chemical, electric, or magnetic stimuli[14–19]. For all the applications, the thermal properties are decisive. For example, as actuators for robotics[20–22] and medical devices, the actuation comes from the LC-isotropic phase transition in macroscopically orientated LCEs (the so-called monodomain LCEs[23,24]). When cooling to a temperature lower than the LC-isotropic phase transition temperature ($T_i$), the monodomain LCEs elongate;

When heated up above $T_i$, they contract[25]. As $T_i$ determines at which temperature the material actuates, $T_i$ is one of the first things needed to consider when choosing the proper LCEs for actuation.

It has been a big challenge since the born of LCEs to tune $T_i$ after LCEs are synthesized. The thermal properties are intimately related to the chemical structures. Usually, those properties have been fixed once the materials are synthesized. To tune those properties, one has to adjust the chemical formulations before the synthesis starts. A possible strategy to change the thermal property is using photo-responsive groups. Some LCEs have photochromic mesogens. When exposed to light, molecular geometric changes, especially trans-cis isomerization, lead to the disappearance of LC phases[16,26]. For LCEs with photo-cross-linkable or photo-polymerizable groups, light can be

[1]The Key Laboratory of Bioorganic Phosphorus Chemistry & Chemical Biology (Ministry of Education), Department of Chemistry, Tsinghua University, Beijing 100084, China. [2]Department of Chemistry, Center for Nanotechnology and Institute of Biomedical Technology, Chung-Yuan Christian University, Chung-Li 32023 Taiwan, China. ✉e-mail: jiyan@mail.tsinghua.edu.cn

used to increase the cross-linking density, which is possible to change $T_i$. However, on the one hand, light has very limited penetration depth; on the other hand, the tuning effect on $T_i$ is very marginal. Annealing is a well-established heat treatment process that can change the microstructure of the material by leaving the material at a certain temperature for some time[27]. Even though it is possible to change $T_i$ by annealing after synthesis due to the structure change during annealing, the new $T_i$ obtained by annealing immediately goes back to the old one when heated to a temperature above $T_i$[28]. For actuation, the LCEs need to go through many heating-cooling cycles around $T_i$. Therefore, annealing is impossible for the previous LCEs to change their actuation temperature. In short, there is no effective strategy to tune the actuation temperature without changing the chemical composition so far.

Here we show that the dynamic covalent bonds and annealing together make it possible to tune $T_i$ after the material is fully covalently cross-linked. Dynamic covalent bonds are capable of reversible bond reformation and breakage based on reversible reactions[29]. The introduction of dynamic covalent bonds results in exchangeable liquid crystalline elastomers (xLCE) which can change their topology while being fully covalently crosslinked[18]. Those kinds of LCEs belong to covalently adaptable networks (CANs)[30,31]. CANs allow the materials to share both the features of thermosets and thermoplastics. For those based on associative exchange reactions, they are also classified as vitrimers[32,33]. Vitrimers can not melt or dissolve, but they can be reprocessed at high temperatures due to accelerated exchange reactions. During the topology change, the cross-linking density remains the same. Besides reprocessablity or recyclability, dynamic covalent bonds also offer LCEs easy preparation of complex shapes and repeated change of actuation modes[34,35]. Moreover, they enable new features such as healing, welding, and reprogramming, which are not available to traditional LCEs[36].

We unintentionally found that the actuation temperatures of LC vitrimers (xLCE-BP) can be strikingly adjusted by annealing at different temperatures. Here, unlike all the previous annealing of LCEs, the new structures obtained by annealing can be fixed thanks to the exchange reaction that occurred during the annealing. Therefore, even though the materials are fully covalently cross-linked and the cross-linking density does not change, the material can possess various $T_i$ without the change of chemical composition. When annealed under external forces, monodomain LCEs form, and the actuation temperatures can be tuned as well. With no need to start from the very beginning of synthesis, one material can be used to meet the different requirements on actuation temperatures for different utilization. This surprising but reasonable strategy not only gives extraordinary convenience for research and application but also is very desirable for a sustainable society.

## Results and discussion
### Synthesis and characterization of xLCE-BP
The model LCE (xLCE-BP) used here was synthesized in a similar way as the previously reported by reacting diglycidyl ether of 4,4'-dihydroxybiphenyl with sebacic acid in the presence of triazobicyclodecene (TBD) as a transesterification catalyst at 170 °C (Fig. 1a)[18]. The TBD in the xLCE-BP was kept at a 0.25% mole ratio to the carboxyl group. The stoichiometry of carboxyl and epoxy groups was 1:1. The detailed preparation procedures of xLCE-BP with different curing times are described in the Supporting Information. The results of Fourier transform infrared (FTIR) spectra clearly showed that the characteristic peaks of the epoxy peak at 909 cm$^{-1}$ disappeared after curing for 3 h (h) (Fig. S1). To eliminate the possible influence of unreacted small molecules on $T_i$, we tried to get rid of the small molecules by swelling. As shown in Fig. S2, the volume does not change after swelling after 6 h. To make sure of full crosslinking and complete removal of small molecules, the xLCE-BP samples used in the subsequent annealing

experiments were cured for 6 h and dried after swelling for 24 h unless otherwise noted. The obtained xLCE-BP is polydomain (no macroscopic orientation and no actuation)[25]. According to the thermogravimetric analysis (TGA) tests, the 5% weight loss of all samples before and after swelling both are around 350 °C (Fig. S3). The catalyst content here is much lower than that used in the previous report which was 5%, but the xLCE-BP here is still a vitrimer that is capable of topology change as the transesterification reaction occurs between the hydroxyl groups and the ester groups[35,37]. Transesterification has a very slow rate at low temperatures, but the reaction rate increases quickly at high temperatures. The material behaves like a traditional thermoset while it is reprocessable at high temperatures.

### Tuning $T_i$ of polydomain xLCE-BP by annealing
$T_i$ of the fully crosslinked xLCE-BP can be easily changed by annealing after the material is synthesized. Differential scanning calorimetry (DSC) was used to trace the change. All the data are acquired on the second heating and cooling procedures with a scan rate of 5 °C/min. The unannealed polydomain xLCE-BP has a glass transition temperature ($T_g$) of 58 °C and a $T_i$ of 114 °C on heating. All the following experiments used this material as the model original material before annealing unless otherwise noted. $T_i$ of the original material before annealing was labeled as $T_{iO}$ and the annealing temperature is labeled as $T_a$ in this work. Unless otherwise noted, the $T_{iO}$ of the xLCE-BP used is 114 °C. When the annealing temperature is lower than $T_{iO}$, the $T_i$ value increases as the annealing time increases. As shown in Fig. 1b, when annealed at 110 °C, on heating, from day 0 to day 15, $T_i$ increases quickly from $T_{iO}$ of 114 °C to 131 °C (3 days), 145 °C (5 days), 155 °C (10 days), and 165 °C (15 days) respectively. After 15 days, the growth of $T_i$ becomes slow and sometimes even drops slightly. However, the overall trend of $T_i$ change is the same. $T_i$ reaches nearly 180 °C after 50 days. On cooling, the $T_i$ change is similar to that of the heating cycles. On both heating and cooling curves, the $T_i$ peak splits as the annealing time extends. Similar splits have been observed in other LC polymers when annealed near $T_i$[38] Such splits were attributed to the polydispersity in the molecular mass. Even though the material here has a network structure, the molecular weight between cross-links has similar polydispersity. The presence of transesterification enables the network to act like linear polymers and the structure rearranged during annealing. The detailed data on $T_i$ changes is shown in Table S1. For the unswollen samples, the value of $T_i$ also changes similarly when annealed (Fig. S4). The detailed data on $T_i$ changes is shown in Table S2.

When the annealing temperature is higher than $T_{iO}$, the opposite trend is obtained, that is, the $T_i$ value of the sample decreases with the increase of annealing time. As shown in Fig. 1c, after the sample was annealed at $T_a = 180$ °C for 3 h, 6 h, 12 h, 24 h, 1 day, 2 days, and 4 days, the $T_i$ value of the samples decreased from $T_{iO}$ of 114 °C to 103 °C (3 hours), 95 °C (6 h), 88 °C (12 h), 81 °C (1 day), 80 °C (2 days) and 77 °C (4 days) respectively during the heating process. As the annealing time increased further, the $T_i$ peak gradually becomes broader and almost overlaps with the glass transition at last. The same trend is observed on the cooling trace.

When we choose the annealing temperature, we consider two factors. One is the $T_i$. Below $T_i$, the material is in the LC phase; above $T_i$, it is in the isotropic phase. According to previous reports of LC materials, annealing below $T_i$ allows the LC phases to grow. The other one is the exchange reaction. The exchange reaction is fast at high temperatures while it is slow at low temperatures. To get a fast exchange rate, we use temperatures as high as possible. When annealing above $T_i$, 180 °C is chosen as annealing at higher temperatures for a long time may lead to oxidation. When annealing below $T_i$, 110 °C is the highest temperature that we choose to ensure fast exchange reactions below $T_i$. Because of the difference in the phase state, the fixed network topology of the material after annealing at

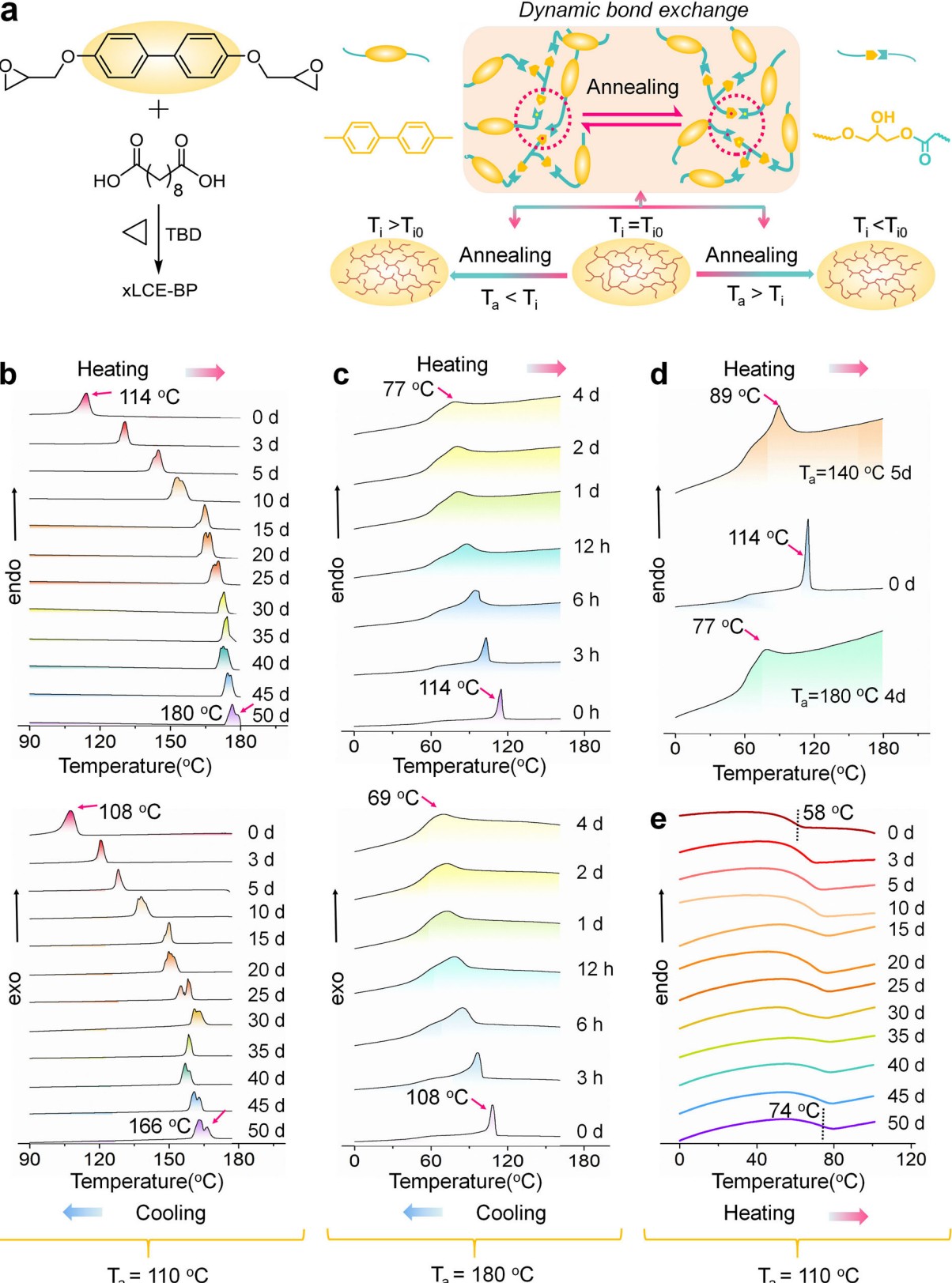

**Fig. 1 | Synthesis of xLCE-BP and tuning $T_i$ by annealing. a** Synthesis of poly-domain xLCE-BP and illustration of tuning $T_i$ by annealing. When the annealing temperature ($T_a$) is above the original $T_i$ value ($T_{i0}$), the new $T_i$ after annealing will be lower than $T_{i0}$. When $T_a < T_i$, the new $T_i$ after annealing will be higher than $T_{i0}$. **b** DSC traces of polydomain xLCE-BP ($T_{i0} = 114$ °C) annealed at $T_a = 110$ °C. **c** DSC traces of polydomain xLCE-BP ($T_{i0} = 114$ °C) annealed at $T_a = 180$ °C. **d** DSC traces of polydomain xLCE-BP ($T_{i0} = 114$ °C) annealed at $T_a = 140$ °C / 180 °C. **e** The change of $T_g$ for polydomain xLCE-BP ($T_{i0} = 114$ °C) annealed at $T_a = 110$ °C. The scanning rates of heating and cooling are both 5 °C/min.

different temperatures is very different, which leads to a great difference in the final $T_i$. The higher the annealing temperature, the faster the decrease is. For example, at $T_a = 140\,°C$ for 5 days, $T_i$ decreased from $T_{i0}$ of $114\,°C$ to $89\,°C$, higher than $77\,°C$ obtained by annealing the same initial xLCE-BP at $180\,°C$ for 4 days (Fig. 1d).

When the annealing temperature is lower than $T_{i0}$, $T_g$ also increases from the original value of $58\,°C$ to $74\,°C$ (Fig. 1e). The detailed data on $T_g$ changes is shown in Table S3. Different non-liquid crystal and non-dynamic covalent polymer systems were synthesized to compare (Fig. S5). Vitrimer-BA and Vitrimer-PU are vitrimers that can change topology due to transesterification and transcarbamoylation reactions, respectively. Polydimethylsiloxane (PDMS) is a common elastomer without dynamic covalent bonds. The synthesis of Vitrimer-BA is the same as the synthesis of xLCE-BP except that the diglycidyl ether of 4,4′-dihydroxybiphenyl was replaced by the diglycidyl ether of bisphenol A[34]. The Vitrimer-PU was synthesized by reacting poly(ethylene glycol)diol (Mn: 400) and glycerin with hexamethylene diisocyanate in the presence of dibutyltin dilaurate (DBTDL) as a catalyst for transcarbamoylation[39]. The PDMS elastomer was prepared from Dow Corning's Sylgard 184 elastomer kit as described in the literature[40]. Detailed synthesis methods for all elastomers are provided in the Supporting Information. All those samples were also swelled for 24 h and dried before the DSC tests. As shown in Fig. S6a, Fig. S6b, the initial $T_g$ of Vitrimer-BA and Vitrimer-PU were $32\,°C$ and $-17\,°C$ respectively. After annealing for 50 days, their $T_g$ did not change significantly, which is very different from the annealing of xLCEs-BP. Similarly, no significantly $T_g$ change was observed in the PDMS system, as shown in Fig. S6c.

## Reversibility and stability of the new $T_i$

Changing $T_i$ of xLCE-BP ($T_{i0} = 114\,°C$) by annealing is reversible as illustrated in Fig. 2a. On the one hand, the sample obtained by annealing below $T_i$ can also show a reduction of $T_i$ value after being re-annealed at $T_a < T_i$. For example, the $T_i$ value of a sample with $T_i$ of $178\,°C$ (which was obtained by annealing the xLCE-BP sample with $T_{i0}$ of $114\,°C$ at $110\,°C$ for 50 days) decreases to $167\,°C$ after being re-annealed at $180\,°C$ for 1 day and to $97\,°C$ after 6 days, respectively. The corresponding DSC results (heating traces) are also shown in Fig. 2b. On the other hand, the $T_i$ value of the sample obtained after annealing at $T_a > T_{i0}$ for different hours can be increased by re-annealing at a temperature lower than its new $T_{i0}$. As shown in Fig. 2c, after being re-annealed at $80\,°C$ ($T_a < T_{i0}$) for 30 days, the $T_i$ of samples annealed at $180\,°C$ for 3 h (the new $T_{i0} = 103\,°C$) and 6 h (the new $T_{i0} = 95\,°C$) rose to $122\,°C$ and $105\,°C$ respectively. Similarly, the value of $T_i$ of those samples annealed at $180\,°C$ for 12 h (the new $T_{i0} = 88\,°C$), 1 day (the new $T_{i0} = 81\,°C$), 2 days (the new $T_{i0} = 80\,°C$), and 4 days (the new $T_{i0} = 77\,°C$) also rose to $92\,°C$, $82\,°C$, $83\,°C$, $79\,°C$ respectively after being re-annealed at $65\,°C$ ($T_a < T_{i0}$) for 50 days. Such an increase is relatively small and slow as the annealing is done at low temperatures, at which the transesterification reaction is slow and the network has low mobility.

The stability of the new $T_i$ is comparable with that of the common xLCEs. It is an in-born shortcoming of vitrimers that the thermal stability is undermined due to fast exchange reactions at high temperatures[41]. To obtain stability actuation, it is better to lower the exchange reaction rate or use other strategies such as getting rid of the catalyst[35,41]. Here, we use low TBD content to lower the transesterification in xLCE-BP. Even though, for the original sample without annealing, $T_i$ drops from $T_{i0} = 114\,°C$ to $83\,°C$ after 35 h of repeated DSC heating-cooling cycles (the final $T_i$ is labeled as $T_{if}$) as shown in Fig. 2d. The stability of samples with high $T_i$ obtained by annealing is very similar to the stability of the $T_i$ of the above original sample without annealing. For example, for the sample with a $T_i$ of $178\,°C$ (obtained by annealing xLCE-BP with $T_{i0}$ of $114\,°C$ at $110\,°C$ for 50 days) $T_i$ gradually decreases and becomes $147\,°C$ after 35 h of the DSC cycle test (Fig. 2e).

This value is still much higher than the unannealed sample ($T_{i0} = 114\,°C$). The stability improves when the $T_i$ is relatively low as the transesterification reaction is slow when the sample is heated to above $T_i$. Even when the temperature is very high, as long as the heating time is not too long, the new $T_i$ is still stable. As shown in Fig. 2f, the sample with a $T_i$ of $77\,°C$ (obtained by annealing xLCE-BP with $T_{i0}$ of $114\,°C$ at $180\,°C$ for 4 days) decreased to $75\,°C$ after 12 h of repeated DSC cycle tests from $-10\,°C$ to $180\,°C$ with a scanning rate of $5\,°C/min$. After continuing the test for 61 h, $T_i$ dropped by only 3 degrees (from $75\,°C$ to $72\,°C$).

The stability indicates that the new structure after annealing is preserved due to the transesterification reaction. Besides the exchange reaction, another possible reason accounting for the structure fixation is the reaction of small molecules or groups. Since the sample after swelling shows $T_i$ change, the structure fixation can not be induced by small molecules. If the fixation were due to the unreacted dangling groups in the network, then, the structure would not show the following reversible $T_i$ change as the reaction between unreacted dangling groups can only happen once.

## The generality of the method

The tuning here can be generalized to LCEs with different chemical compositions. We synthesized three more LCEs with dynamic covalent bonds according to the previous report (Fig. S7). The synthesis of xLCE-DHMS is the same as the synthesis of xLCE-BP except that the diglycidyl ether of 4, 4′-dihydroxybiphenyl was replaced by the diglycidyl ether of 4,4′-dihydroxy-methylstilbene[34,42]. xLCE-PU samples were prepared by a classic "one-pot" reaction between hydroxyl and isocyanate[7]. xLCE-RM257 was synthesized referring to our previous work[43]. Detailed synthesis is provided in the Supporting Information. Those materials were also swelled and dried before annealing. According to previous reports, xLCE-DHMS has a smectic phase while xLCE-PU and xLCE-RM257 have nematic phases. No matter whether smectic or nematic, the obtained materials are all polydomain LCEs. On heating, the value of $T_{i0}$ for xLCE-DHMS, xLCE-PU, and xLCE-RM257 is $84\,°C$, $102\,°C$, and $88\,°C$ respectively, and on cooling, the value of $T_{i0}$ for xLCE-DHMS, xLCE-PU, and xLCE-RM257 is $81\,°C$, $97\,°C$, and $79\,°C$ respectively. They all show a similar $T_i$ change upon annealing. For example, when annealed at $80\,°C$ ($T_a < T_{i0}$) for 50d, the $T_i$ value of the xLCE-DHMS increased from $T_{i0}$ of $84\,°C$ to $102\,°C$, and xLCE-PU increased from $T_{i0}$ of $102\,°C$ to $119\,°C$ on heating. (Fig. S8a, b). On cooling, after annealing at $80\,°C$ ($T_a < T_{i0}$) for 50d, the $T_i$ value of the xLCE-DHMS increased from $T_{i0}$ of $81\,°C$ to $95\,°C$ and xLCE-PU increased from $T_{i0}$ of $96\,°C$ to $111\,°C$. For xLCE-DHMS ($T_{i0} = 81\,°C$), the choice of annealing temperature is similar to the annealing xLCE-BP ($T_{i0} = 114\,°C$) at $110\,°C$, that is, $T_a$ is lower than $T_{i0}$ while $T_a$ is very close to $T_{i0}$. For xLCE-PU, the annealing temperature is much lower than $T_{i0}$, but the conclusion that annealing at $T_a < T_{i0}$ increase $T_i$ still stand. However, the increment of $T_i$ for xLCE-DHMS and xLCE-PU is much smaller than that of xLCE-BP. Similar to the annealing of xLCE-BP at temperatures above $T_{i0}$, the $T_i$ value decreases. For one example, when xLCE-RM257 was annealed at $80\,°C$ (slightly higher than the $T_{i0}$ of xLCE-RM257 measured from the DSC cooling trace), as shown in Fig. S8c, the value of $T_i$ decreases, and after 20 days, $T_i$ and $T_g$ completely overlap. This experiment proves that annealing at a temperature slightly above $T_{i0}$ can also decrease $T_i$. For another example, the $T_i$ value of xLCE-DHMS decreases from $84\,°C$ to $66\,°C$ after annealing at $140\,°C$ for 5 days and $55\,°C$ after annealing at $180\,°C$ for 2 days Fig. S9 respectively. This experiment also proves that annealing at a temperature above $T_{i0}$ can not only decrease $T_i$, but also the higher the temperature, the faster the rate of declination. The detailed data of DSC for three liquid crystal elastomers annealed at $80\,°C$ are shown in Table S4. As we can see from the three LCEs annealed at $80\,°C$, even though the same annealing temperature is used, the change of $T_i$ varies as the molecular structure changes. More systematic work is necessary

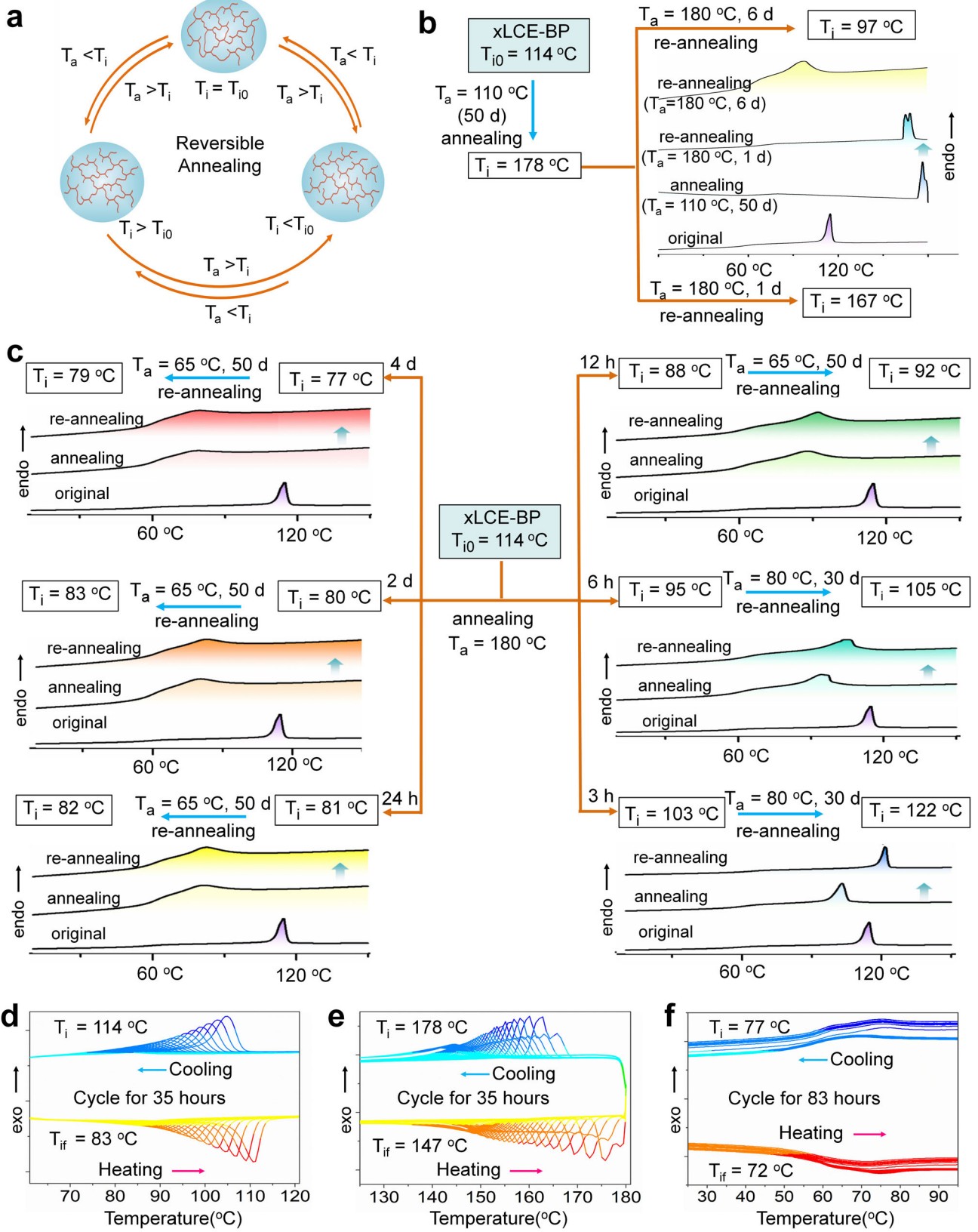

**Fig. 2 | Reversibility and stability of $T_i$ obtained by annealing. a** The illustration on the reversibility of $T_i$ by re-annealing. **b** Re-annealing ($T_a > T_i$) of the samples obtained by annealing polydomain xLCE-BP ($T_{i0} = 114\,°C$). **c** Re-annealing ($T_a < T_i$) of the samples obtained by annealing polydomain xLCE-BP ($T_{i0} = 114\,°C$). **d** Thermal cycle tests of polydomain xLCE-BP without annealing ($T_i = 114\,°C$). **e** Thermal cycle tests of polydomain xLCE-BP ($T_i = 178\,°C$) obtained by annealing xLCE-BP ($T_{i0} = 114\,°C$) at 110 °C for 50 days. **f** Thermal cycle tests of polydomain xLCE-BP ($T_i = 77\,°C$) obtained by annealing xLCE-BP ($T_{i0} = 114\,°C$) at 180 °C for 4 days. The scanning rates of heating and cooling are both 5 °C/min.

to further clarify the effect of exchange reaction, molecular structure and phase behavior on the tuning of $T_i$ by annealing.

## Tuning the actuation temperature of monodomain xLCE-BP

When annealing is done while the sample is being stretched, we can get monodomain xLCE-BP. Polymers without dynamic covalent bond system can not fix the structures obtained by annealing, so its $T_i$ immediate drops to $T_{i0}$ after being heated to a temperature higher than the $T_i$ obtained by annealing. Unlike traditional LCEs, the liquid crystal elastomers with dynamic covalent bonds can fix the new network topology, so that the obtained new $T_i$ does not disappear after being heated to above $T_i$. The difference between xLCE and other polymers is as illustrated in Fig. 3a. xLCE gives out an opportunity to tune the actuation temperature of monodomain LCEs.

Similar to the polydomain samples, different annealing time at different temperature gives out monodomain xLCE-BPs with different $T_i$. The details of the monodomain preparation can be found in the Supporting Information. As shown in Fig. 3b, in the first 15 days, on heating, the $T_i$ of the monodomain samples increased gradually to 133 °C (3 days), 137 °C (5 days), 153 °C (10 days), and 161 °C (15 days), respectively; On cooling, the $T_i$ values of sample are 122 °C (3 days), 124 °C (5 days), 134 °C (10 days) and 140 °C (15 days), respectively. The detailed data on $T_i$ changes is shown in Table S5. The splits of the $T_i$ peak can also be observed on heating and cooling. Compared to polydomain samples, such a split gives out more sub-peaks. Probably the alignment of the mesogens further promoted the polymorphism[44]. Much more work is necessary to further clarify the effect of alignment on the splitting of the peaks.

The actuation shares a similar thermal stability to the stability of $T_i$. The actuation of the samples with low $T_i$ is very stable because the transesterification rate is slow at low temperatures. Many actuation cycles can be completed while the actuation strain remains unchanged. For example, as shown in Fig. 3c, the sample with $T_i$ of 90 °C obtained by annealing at 180 °C for 24 h still has good stability after 100 h of the DMA test with 100 heating-cooling cycles. However, with the increase of $T_i$, the stability of the actuation decreases due to the rapid transesterification reaction at high temperatures, which is the same as the properties of xLCE with high $T_i$ reported previously[35,41]. In this work, the catalyst content is low. So, the samples with high $T_i$ still have certain actuation stability. For example, even after 200 rapid heating-cooling cycles on a hot plate, the actuation strain of the sample with a $T_i$ of 150 °C hardly changed (Fig. 3d). In the above 200 cycles, it takes about 7 min to complete a contraction-elongation cycle between 140 °C and 160 °C. The heating process takes about 2 min and the cooling process takes about 5 min. This contraction-elongation cycle is carried out for 200 times in total (one cycle is shown in Supplementary Movie 1).

Based on the $T_i$ change, instead of starting from synthesis, the fully cross-linked xLCE-BP can perform actuation at different temperatures with notable actuation stability. Figure 3e shows five monodomain xLCE-BP films with $T_i$ values of 90 °C, 112 °C, 122 °C, 132 °C, and 150 °C, respectively. The sample with $T_i$ of 90 °C ($T_{i1}$) and 112 °C ($T_{i2}$) were obtained by annealing polydomain xLCE-BP ($T_{i0}$ = 114 °C) after stretching at 180 °C for 24 h and 3 h, respectively. The samples with $T_i$ of 122 °C ($T_{i3}$), 135 °C ($T_{i4}$) and 152 °C ($T_{i5}$) were obtained by annealing xLCE-BP ($T_{i0}$ = 114 °C) at 110 °C after stretching for 3 d, 10 d and 30 d, respectively. The five samples are placed together. When the temperature goes to 100 °C, only the sample with a $T_i$ of 90 °C contracted. When the temperature continues to go up, at 120 °C, 130 °C, 140 °C, and 160 °C, the samples with $T_i$ of 112 °C, 122 °C, 135 °C, and 152 °C, contracted one by one.

## Investigation of the structure change during annealing by X-ray diffraction(XRD)

To understand the changes in molecular structure by annealing, XRD tests were performed for the xLCE-BP samples obtained by annealing at

different conditions. 2D XRD images were obtained on Ganesha system (SAXSLAB) equipped with a multilayer focused Cu Kα radiation as the X-ray source (Genix 3D Cu ULD) and a 2D semiconductor detector (Pilatus 300 K, DECTRIS, Swiss). All the xLCE-BP samples exhibit the characteristic X-ray pattern of smectic A phases. For the original polydomain xLCE-BP without thermal treatment ($T_{i0}$ = 114 °C), there are two inner rings with a spacing of 29.50 Å (2θ = 2.99°) and 14.88 Å (2θ = 5.93°), respectively. Meanwhile, there is an outer ring with a spacing of around 4.38 Å (2θ = 20.25°), as shown in Fig. 4a. According to Fig. 4a, no matter whether the sample was annealed below or above $T_i$, the resulted polydomain xLCE-BP samples after annealing still have the lamellar structure of smectic A phase. It is obvious that when annealed above $T_i$, the intensity of the inner rings greatly becomes weak, and some rings even disappear. According too the results of 1D XRD derived from the 2D XRD data (Fig. 4b), the signal peaks of 2θ = 2.99° and 5.93° shifts to the right after annealing at 110 °C ($T_a < T_{i0}$), which means that 2θ increases and the d-spacing between layers becomes smaller (Fig. S10). On the contrary, when $T_a > T_{i0}$ the intensity of the signal peaks around 2θ = 2.99° and 5.93° greatly decreased. For example, as shown in Fig. 4c, after being annealed at 180 °C for 4 days, the peak at 2θ = 3.15° (d = 28.05 Å) almost disappeared, and the peak at 2θ value of 6.11° (d = 14.44 Å) becomes weaker. A similar phenomenon is found in the sample annealed at 140 °C for 5 days, but the magnitude of intensity decrease is less than that of samples annealed at 180 °C in Fig. 4c. Therefore, annealing changes the structure of the network. Annealing at $T_a < T_{i0}$ helps to the formation of a relatively more compact layered structure while annealing at $T_a > T_{i0}$ impairs the regularity of lamellar stacking as illustrated in Fig. S11.

XRD diffraction experiments also proved the successful preparation of monodomains (Figs. 4d, e). As shown in Fig. 4d, all 2D XRD patterns show a series of peaks as indicated by sharp inner reflections on the meridian instead of rings as observed in the 2D XRD pattern of polydomain samples. Compared to the polydomain samples, there are two additional peaks observed at about 2θ = 9.0° and 12.0° as shown in Fig. 5e when annealed at 110 °C ($T_a < T_{i0}$). The diffraction peaks located around 2θ = 2.99° and 5.93° are consistent with those of the polydomain samples with a d-spacing of around 29.50 Å and 14.88 Å respectively. However, as observed from the derived 1D X-ray data, both peaks split after annealing at 110 °C as shown in Fig. 4e. For example, for the sample annealed at 110 °C for 3 days, the peak around 29.50 Å in the original sample split into 29.05 Å and 32.86 Å. In 15 days, those peaks are at 27.23 Å and 29.00 Å respectively. Such splits also confirm the possible coexisting of different smectic A layers, which have been indicated by the splits of the $T_i$ peak on DSC traces. The d-spacing decreasing in the first 5 days. After 5 days, the splits still exist, but there is no clear decreasing or increasing trend.

In consist with the unannealed samples, for the monodomain xLCE-BP samples annealed at $T_a > T_{i0}$ the reflections at 2θ = 2.99° and 5.93° are greatly weakened and sometimes are hard to discern. For example, as shown in Fig. 4e (top) the sample annealed at 180 °C for 24 h has only 3 diffraction peaks. Although it is still a lamellar structure, the diffraction peaks with 2θ value of 2.99° (d = 29.50 Å) and 9.01° (d = 9.81 Å) both disappear. The intensity of the remaining diffraction peaks also becomes much lower. Detailed XRD results of polydomain and monodomain were shown in Table S6 and S7. The variation of structure with annealing temperature and time not only appeared in xLCE-BP but was also found in xLCE-DHMS. Detailed 2D XRD results of xLCE-DHMS annealed at $T_a > T_{i0}$ are shown in Fig. S12 and Table S8.

## Patterning different $T_i$ on the same film

The change of $T_i$ helps to expand the application of LCEs. For example, using digital patterning and electrothermal films, different $T_i$ can be patterned on the same liquid crystal elastomer film as illustrated in Fig. 5a, which can be applied to anti-counterfeiting. Patterning different $T_i$ in the same LCE film was not possible in the past without

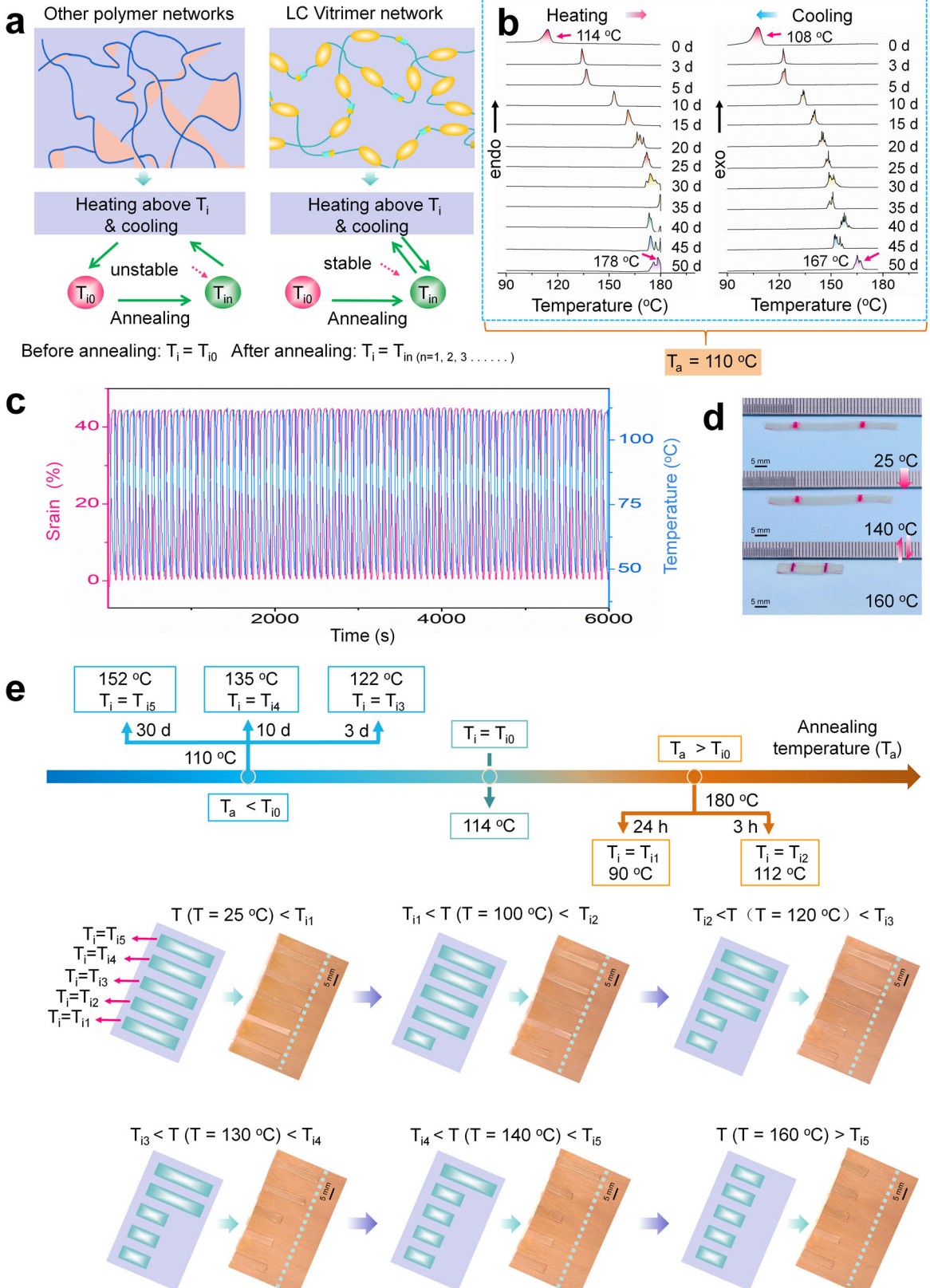

**Fig. 3 | Tuning the acuation temperature of monodoamin xLCE-BP. a** The difference between LC vitrimer networks and other polymer networks on the stability of $T_i$ obtained by annealing. **b** DSC traces of monodomain xLCE-BP after annealing below $T_i$. The scanning rates of heating and cooling are both 5 °C/min. **c** The contraction-elongation actuation of the aligned xLCE-BP for 100 heating-cooling cycles. **d** Thermal actuation of an aligned xLCE between 160 °C (isotropic phase) and 140 °C (liquid-crystal phase) after 200 rapid heating-cooling cycles on a hot plate. **e** Fabrication of five monodomain materials with different actuation temperatures and demonstration of five monodomain actuators actuate at different temperatures.

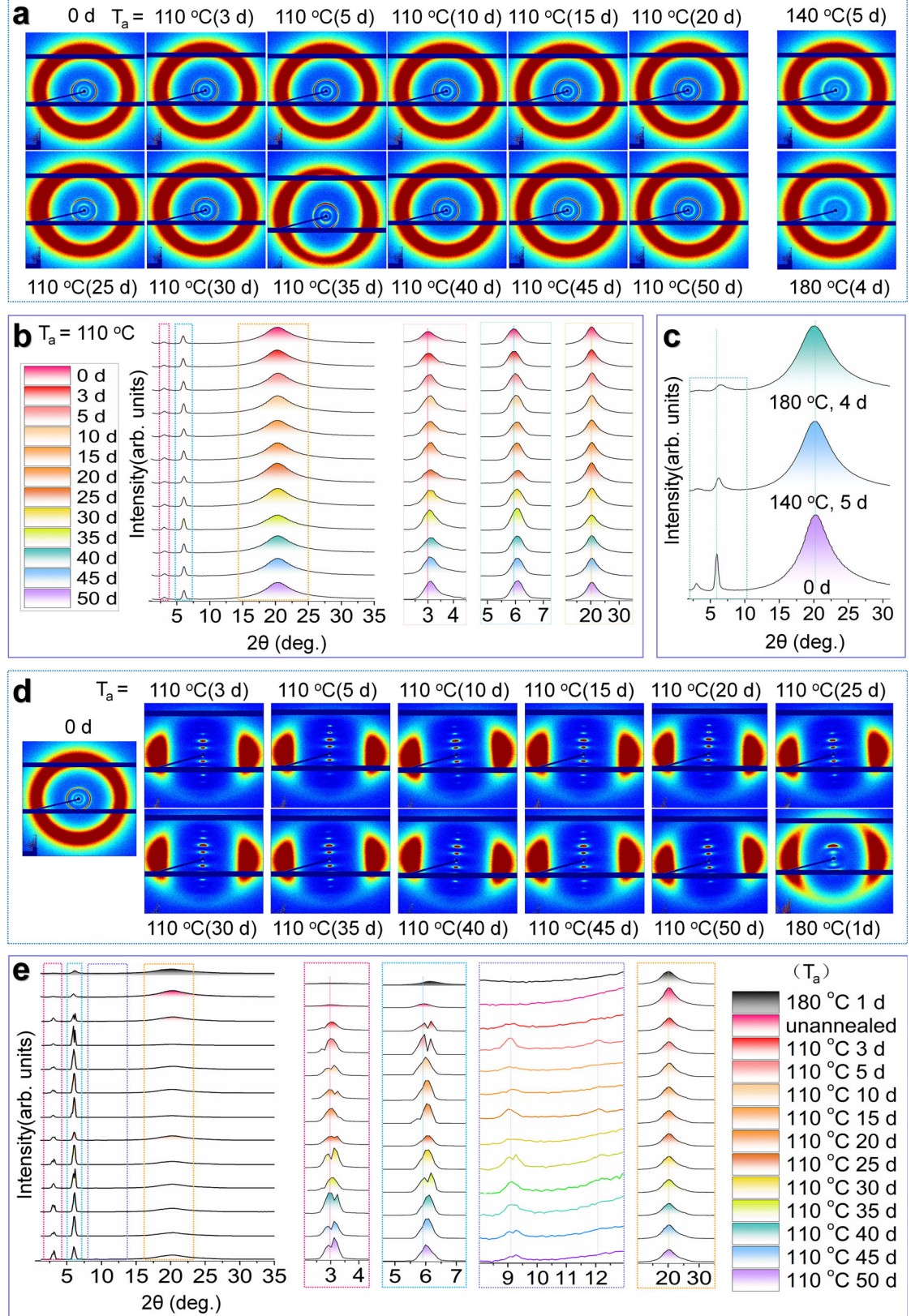

**Fig. 4 | Tracking the structure changed during annealing by X-ray diffraction.**
**a** The 2D X-ray diffraction patterns of unannealed polydomain xLCE-BP sample
($T_{iO} = 114\,°C$) and those after annealing at different temperature for different time.
**b** Integrated 1D X-ray diffraction profiles of the unannealed polydomain xLCE-BP
($T_{iO} = 114\,°C$) and the samples annealed at 110 °C ($T_a < T_{iO}$). **c** Integrated 1D X-ray
diffraction profiles of the unannealed polydomain xLCE-BP ($T_{iO} = 114\,°C$) and the

samples annealed at 140 °C and 180 °C ($T_a > T_{iO}$). **d** The 2D X-ray diffraction pat-
terns of monodomain xLCE-BP obtained by annealing stretched polydomain xLCE-
BP ($T_{iO} = 114\,°C$) at different temperature for different time. **e** Integrated 1D X-ray
diffraction profiles of the monodomain xLCE-BP obtained by annealing stretched
polydomain xLCE-BP ($T_{iO} = 114\,°C$) at different temperature for different time and
the unannealed polydomain xLCE-BP ($T_{iO} = 114\,°C$).

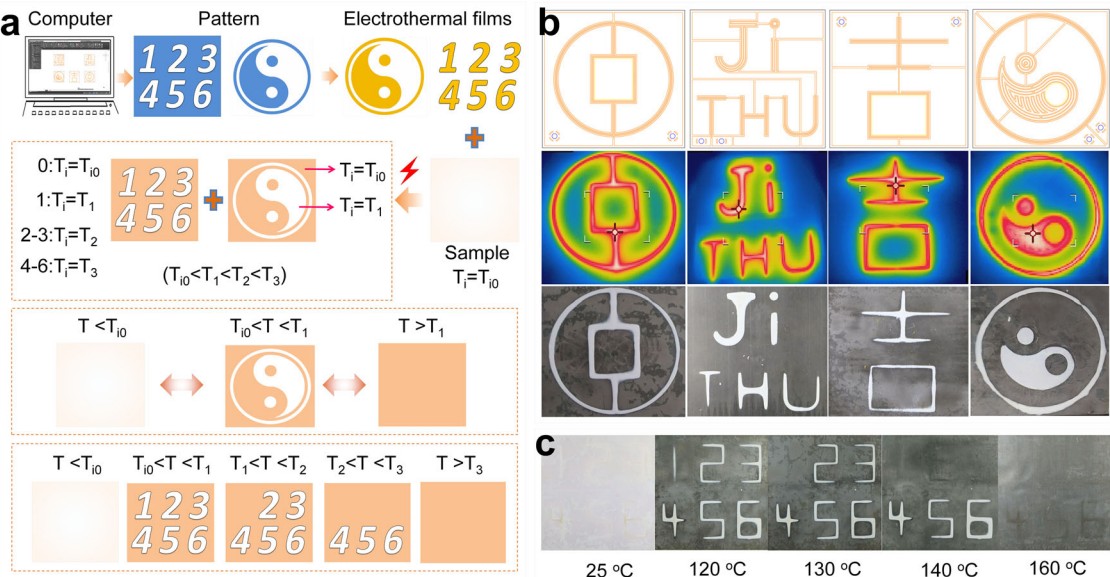

**Fig. 5 | Patterning different $T_i$ on the same film. a** Illustration on the process of patterning different $T_i$ on the same film. Different patterns were first designed by computer. Then the electrothermal PI films were made into those patterns. An xLCE film was attached to the PI films. When heating different areas for different time, different areas obtained different $T_i$. When the temperature changes, the pattern changes. **b** Patterns of polydomain xLCE-BP ($T_{i0} = 113\,°C$) with dual $T_i$ values. **c** Patterns of polydomain xLCE-BP ($T_{i0} = 113\,°C$) with multiple $T_i$ values.

assembly. As shown in Fig. 5b, various patterns were designed through CAD software and used to make the corresponding electrothermal polyimide (PI) films with the same patterns.

When the PI film is attached to the polydomain xLCE-BP ($T_{i0} = 114\,°C$) and applied voltage, the patterned area is heated to the required temperature as shown in Fig. 5b (middle row). The $T_i$ value of unheated part remains 113 °C while the $T_i$ value of the patterned part increases to 135 °C. When cooling down, the whole LCE film becomes opaque. As the temperature rises to 120 °C, the unheated part becomes transparent as it is now in the isotropic phase. The patterns appeared since the patterned part is still opaque since it is in the LC phase, as shown in Fig. 5b (bottom raw). By controlling the annealing time, there can also be more than two kinds of $T_i$ values on the same film. As shown in Fig. 5c, the unheated part has a $T_i$ is 113 °C, the $T_i$ value of number "1" is 125 °C, the $T_i$ values of "2" and "3" are both 135 °C, and the $T_i$ values of "4", "5", and "6" are all 152 °C. At 120 °C, "1 ~ 6" all are visible, At 130 °C, "1" disappeared. At 140 °C, "2" and "3" are invisible. At 160 °C, all the patterns disappear.

In this work, without changing the chemical composition of the material, the $T_i$ value of the fully cross-linked liquid crystal networks can be reversibly adjusted by annealing at different temperatures for different time. In contrast to the immediate high-temperature disappearance of the structure obtained by annealing of LCs without dynamic covalent bonds, the annealed structure is preserved by the exchange reactions in the liquid crystal elastomers with dynamic covalent bonds. The stable new $T_i$ guarantees the actuation at new temperatures. Therefore the same material with exactly the same chemical composition can be used as actuators to meet different actuation temperature requirements. When properly designed, the strategy here will allow the same structure made of one single material to have different local actuation temperatures, and enable complex actuation mode. $T_i$ change also means the changes in mechanical and other properties, which will extend the application of liquid crystal elastomers.

## Data availability
The data generated in this study have been deposited in the Science Data Banke database under the link address. https://cstr.cn/31253.11.sciencedb.08266.[45] Source data are provided with this paper.

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

## Acknowledgements

This work was supported by the National Natural Science Foundation of China (Nos. 51722303(Y.J.), 21674057(Y.J.), and 21788102(Y.W.)).

## Author contributions

Y. Yao found the changing of $T_i$ by annealing xLCE. Y.J. and Y. Yao developed the idea. Y. Yao performed the experiments. Y. Ji and Y. Yao wrote the draft. E. He helped to measure DSC, H. Xu provided xLCE-RM257 samples, Y. Liu provided PDMS samples, Z. Yang helped to revise and check the manuscript, and Professor Y.W. provided part of the financial support.

## Competing interests

The authors declare no competing interests.
