## [Peer Review File · Nature Communications]

REVIEWER COMMENTS

Reviewer #1 (Remarks to the Author):

The paper is interesting and presents nice results about the activation temperature modulation in LCE with dynamic covalent bonds. The novelty is surely in the stability of the new isotropization temperature achieved by annealing, while the synthetic approaches and xLCE were already extensively reported (and cited within the paper).

The first problem I found was about the clarity of the paper (comprising figures and the supporting information). The language must be strongly revised, many typos and repetition make the reading hard, and the paper is chaotic forcing the reader to go back and forth through the paper/SI pages and looking for the needed information from SI, texts and figures continuously. Some examples:

- "casual" order of the cited SI figures: S3 is the first cited followed by S5, then S1, etc.
- DSC traces do not report which (cooling or heating?) cycle are representing and if exo-up or endo-up representation was chosen. In the same figure different DSC traces reports opposite sequences
- Explanation about DSC peak splitting is only done after many pages together with XRD discussion
- The same figure (e.g. S1) is described in very different sections of the paper
- Figures are in general not self-explicative and in some case, it is very difficult to understand the starting Ti or the used temperatures
- Figure S2, S4 and S8 and table S1 and S4 are not cited within the text.

About the scientific part:

In general a more systematic approach should be adopted, for example by using the same annealing temperature for all samples investigated

Introduction:

I believe better references could be selected for the general LCE description. Many reviews were reported on LCE applications in robotics, micromachines, artificial muscles and tissue engineering, on their response upon thermal, optical, chemical, electric or magnetic stimuli, and on LCE containing photochromic mesogens and the mechanisms behind light actuation, better describing the current state of the art (to mention a few, see reviews from: Schenning, Broer, Wiersma, Heggman, Ware, White, Ikeda, Priimagi, Yu, etc. group).

Tuning T_i of polydomain xLCE-BP by annealing:

Nor in the text neither in Figure 1 is reported the temperature used for annealing below T_i (110°C?), while the initial T_i of the sample is mentioned within the text only at the end of figure 1d description (reporting the data at the beginning of the description would help the reader understanding the data). The same problem is present for figure 1e: annealing temperature? Since the annealing temperature is fundamental to modulate the T_i of the material, it should be clear both within the text and the figures the temperature used and reached (in terms of T_i) by the samples.

Why such a great difference in annealing below and above the T_i ? Below the T_i a quite close T is chosen (110 °C is quite close to 114 °C) while for annealing above a T_i much higher difference in temperature is used (180 °C is 66 °C higher than T_i). Which is the scientific explanation for this difference?

Figure S1: it would be beneficial to also have the structure or a representation of the final products

Description of Figures S6c and d in the text are exchanged (Figure S6c reported data from Vitrimer-PU not Vitrimer-BA)

DSC traces reported in Figure S6b cannot be used to determine the T_g : no peaks are evident from the reported traces, a DMA analysis will be more correct in this case.

Stability and the reversibility of the new T_i .

Figure 2b: what happens after 12h? Longer experiments were done?

Even in this section reporting in the text and in the Figure explicating the temperatures mentioned would help the reader.

The generality of the method

This section does not explicitly report any T . From the supporting I gain the following:

xLCE-DHMS $T_i=80/84^\circ\text{C}$ (heating/cooling)

xLCE-PU $T_i=97/102^\circ\text{C}$ (heating/cooling)

xLCE-RM257 $T_i=79/88^\circ\text{C}$ (heating/cooling)

but these temperatures are referred to polydomain samples, while description is reported for xLCE-DHMS in the smectic phase and xLCE-PU and xLCE-RM257 in the nematic phase. The author should explicit the temperatures within the text and explain to a broader readership that these temperature does not change in this case.

From table S3 I understood that all annealing experiments for the 3 samples were performed at 80°C, considering 80°C a lower T (with respect to T_i) for xLCE-DHMS and xLCE-PU and a higher T (with respect to T_i) for xLCE-RM257.

Why a so close (at least for xLCE-DHMS and xLCE-RM257) T was chosen? Would not be more correct to select at least 75°C and 85°C as lower or higher temperatures to have some degrees in between the annealing T and the Ti? Why experiment comparable to the previous ones, therefore also performed at 65°C, 110°C and 180°C were not repeated?

Figure S7 reports different sections and scales for the 3 traces, it would be better to have all the scales from 60°C to 180°C with the same scale.

Preparation and tuning of the actuation temperature of monodomain xLCE-BP

Figure 3b is partially describe before and after figure 3c. It would be better to describe the figure all at the same point (before figure 3c). Moreover, the pictures on the right of the boxes are not well defined, maybe a higher contrast will help seeing the differences within the LCE strips.

The authors wrote: "The splits of the Ti peak can be observed on heating and cooling. Compared to polydomain samples, such a split gives out more sub-peaks." Without explain and discuss this phenomenon. Please discuss this point.

Furthermore, relating to the phrase: "even after 200 rapid heating-cooling cycles on a hot plate, the actuation strain of the sample with a Ti of 150°C hardly changed (Figure 3e)" what is considered "rapid"? times? Please better detail, explain and discuss this aspect!

About Figure 3: temperatures and behavior at 0d should be make explicit in Figure 3c and 3b, respectively. Panel e is too small to be understandable and scales are missing.

Investigation of the structure change during annealing by XRD

Also in this case is very difficult to understand the annealing conditions of the data reported!

The author wrote: "Figure 4b shows the dependence of the lamellar spacing on annealing time at respective temperatures." Where are these temperatures indicated? I cannot find them nor in the text neither in Figure 4b.

Furthermore: "In the first 15 days, the spacing decreases as the annealing time is increased and reaches 28.15 Å on the 15th day." It looks to me that this is true after 5 and not 15 days.

"It looks like that annealing lower than Ti helps to the formation of a relatively more compact layered structure, but loner time annealing may not be good." Also in this case, annealing conditions? Ti? Where this data are reported?

About the comparison of the experiments performed at 110°C, 180°C and 140°C why they do not report the correspond data also at the same time?

Figure 4f only describe the data recorded after annealing at 180°C for 24h and not the data after annealing 3d at 110°C. Moreover, the description reported looks more appropriate to the sample annealed at 110°C for 3 days.

Reviewer #2 (Remarks to the Author):

This is an exciting study demonstrating a simple and general strategy for tuning the isotropization temperature (T_i) for liquid crystalline elastomers (LCEs). T_i is a very important parameter for LCEs as it determines the actuation temperature and related properties for LCE functional applications. The traditional methods for changing T_i of a LCE is through modifying its chemical structure, which is tedious and not general. While physical methods like annealing can induce the change of T_i , one major issue is that the new T_i after annealing immediately goes back to the old one when heated above the T_i , which limits its practical applicability. As the authors pointed out in the introduction, there is no effective strategy to tune the actuation temperature without changing the chemical composition so far. In this study, the authors made a serendipitous discovery demonstrating that the dynamic covalent bonds and annealing together make it possible to tune T_i after the material is fully covalently crosslinked. The key to achieve this tunability of T_i is through the use of dynamic covalent crosslinks to the LCEs. In nutshell, the tunability was achieved by taking advantage of two synergistic thermal effects: 1) thermal annealing to tune actuation temperature, and 2) dynamic covalent bond exchange during annealing and subsequent fixing upon removal of the heat. In contrast to normal LCEs that lose new T_i after heating up the annealed samples, in the new systems reported here different T_i 's achieved by annealing can be preserved by reversible reaction of dynamic covalent bonds in covalently adaptable LC networks including LC vitrimers. The authors further demonstrated the generality of this approach by showing similar behavior in several LCE systems using different dynamic covalent bonds including transesterification, urethane exchange, etc. The authors did an excellent job in showing the tunability of T_i over a broad range by simple annealing. They also showed convincingly that the annealed samples show good stability for the new actuation temperature over many heating/cooling cycles. The structural change during annealing of the LCE samples were carefully monitored by XRD. Lastly, the authors creatively showed patterning on LCE film by precisely controlling annealing using digital patterning and electrothermal films. Overall, this work demonstrates an exciting new and general methods for tuning the actuation temperature of LCE materials, which makes an important contribution to the dynamic polymer materials area. This reviewer would enthusiastically support the acceptance of this work by Nature Communication after addressing the following minor issues.

- In the discussion, the authors should make it more clear how physical annealing changes the actuation temperature for LCEs. The authors cited some references in the manuscript for this effect but it would help readers a lot by briefly discussing how annealing at different temperature changes the structure differently and accordingly the actuation temperature of LCEs.

- Some figures/data are a little hard to follow which need some revision/improvement. For example, Fig. 3b, it takes multiple reading to figure out the experiments. Perhaps label T1, T2, T3,... next to each sample.

- Some writing expressions need to be revised. For example, pp 3, line 6: "this highly bright promise" → "this promise". Pp18, line 8: "In Figure 3b shows" → "Figure 3b shows".

Reviewer #3 (Remarks to the Author):

The manuscript identifies a new methodology for tuning the nematic to isotropic phase transition temperature in isotropic-genesis and mechanically aligned LCE materials containing covalent adaptable networks (xLCEs). The methodology enables the transition temperature to be tuned in a relatively wide range of temperatures. The authors provide evidence of their deterministic ability to tune the temperature to their desired values across a range of xLCE materials and confirm their results using a combination of experimental methods, the main of which are DSC and XRD.

To the best of this reviewer's knowledge, the results presented in this work are original, the used chemical compositions and experimental methodologies are scientifically sound and the obtained results are correct, new, interesting, and complement the contemporary literature in the field. I, therefore, recommend the publication of this manuscript in Nature Communications with the following comments:

1. Significant changes to the shape of the Ti peak (involving phenomena such as broadening and splitting), as well as significant changes to the Tg (along one direction of the annealing process) were observed throughout the first section of the manuscript. The authors were further able to demonstrate that their methodology does not apply to non-LC and non-CAN materials. However, although the reversibility of the process is announced, the changes in peak shape indicate that the material does change in certain ways (e.g., Figure 2f). It is my opinion that the manuscript would benefit from an additional discussion of this issue.

2. The XRD results do provide the necessary evidence for the statements that have been made. However, in figures 4e and 4f, the specific choice of annealing temperatures (110C and 180C) is somewhat unclear. A short explanation qualifying these choices of evaluation temperatures in terms of their effects on Ti and their comparability would do well to strengthen the authors' point.

3. The main experimental details of the XRD method should be stated directly in the main text and not only in the supplementary. Figure 4 would benefit from adding a scalebar for the reciprocal space or some notation for the obtained reflections.

5. The authors are encouraged to perform an additional language proof to bring the writing up to the journal's standards.

Point-to-point response to the reviewers' comments

Title: Enabling liquid crystal elastomers with tunable actuation temperature

Manuscript ID: NCOMMS-22-53652

First of all, we are grateful to all reviewers for carefully reading the manuscript and for many constructive suggestions that lead to essential improvements in our paper. We have tried our best to revise this article according to the comments of the reviewers. A point-to-point response is provided in the following. We also highlighted the changes made in our "Revised manuscript" and "Revised Supplementary Information" with a yellow background in the "Combined PDF" document.

(Reviewers' comments: in black; **Corrections made by the authors in response to the comments: in red; What we rewrote in the revised manuscript: in red and underlined**)

REVIEWER COMMENTS

Reviewer #1 (Remarks to the Author):

The paper is interesting and presents nice results about the activation temperature modulation in LCE with dynamic covalent bonds. The novelty is surely in the stability of the new isotropization temperature achieved by annealing, while the synthetic approaches and xLCE were already extensively reported (and cited within the paper).

Reply: We thank the reviewer very much for the positive comment on our work. We also appreciate the concerns that the reviewer raised, which helped us a lot to improve the work.

Comment 1:

The first problem I found was about the clarity of the paper (comprising figures and the supporting information). The language must be strongly revised, many typos and repetition make the reading hard, and the paper is chaotic forcing the reader to go back and forth through the paper/SI pages and looking for the needed information from SI, texts and figures continuously. Some examples:

Reply 1: We are very sorry for the bad clarity due to our awkward writing skills, which must have caused much trouble to the reviewer and made the reviewer spend a lot of extra time to understand our work. We thank the reviewer very much for the detailed advice on how to improve the clarity of this paper. Following the reviewer's advice, we have made the corresponding changes.

Comment 1.1:

"casual" order of the cited SI figures: S3 is the first cited followed by S5, then S1, etc.

Reply 1.2: According to the reviewer's comment, all the SI figures and tables cited in the paper have been adjusted in order in the "Revised manuscript" and the "Revised Supplementary Information".

Comment 1.2:

- DSC traces do not report which (cooling or heating?) cycle are representing and if exo-up or endo-up representation was chosen. In the same figure different DSC traces reports opposite sequences.

Reply 1.2: Following the reviewer's advice, the marks of the heating (Red arrow) and cooling (Blue arrow) have been added to all DSC figures in the "Revised manuscript" and "Revised Supplementary Information". We also added "endo" and "exo" to all DSC tests.

Comment 1.3:

- Explanation about DSC peak splitting is only done after many pages together with XRD discussion

Reply 1.3: We have shifted the explanation about DSC peak splitting forward to the part following the description about DSC traces.

Comment 1.4:

- The same figure (e.g. S1) is described in very different sections of the paper

Reply 1.4: In the revision, Figure S1 is divided into two figures: Figure S5 and Figure S7 respectively, and rearranged according to their order in the text. Throughout the manuscript, we tried to describe the same figure in the same section.

Comment 1.5:

- Figures are in general not self-explicative and in some cases, it is very difficult to understand the starting T_i or the used temperatures

Reply 1.5: Following the reviewer's advice, we have tried our best to re-do all the figures. We also marked all the "starting T_i " as " T_{i0} ", the annealing temperature as " T_a " and the final temperature after annealing as " T_{if} ".

Comment 1.6:

- Figure S2, S4 and S8 and table S1 and S4 are not cited within the text.

Reply 1.6: All figures and tables in the "Revised Supplementary Information" have been reorganized and cited in order in the "Revised manuscript".

Comment 2:

About the scientific part:

In general, a more systematic approach should be adopted, for example by using the same annealing temperature for all samples investigated

Reply 2: We do agree with the reviewer that using the same temperature for all samples is more systematic usually. However, here, it is not easy to adopt the same annealing temperature. Annealing below T_i and annealing above T_i is different. The value of T_i of different samples varies greatly, so we can not use the same temperature for different materials. Besides, the exchange reaction in different materials is also different. Moreover, for different materials, the range of T_i changes after annealing is also different. Therefore, annealing temperatures can only be selected according to specific materials.

Comment 2.1:

Introduction:

I believe better references could be selected for the general LCE description. Many reviews were reported on LCE applications in robotics, micromachines, artificial muscles, and tissue engineering, on their response to thermal, optical, chemical, electric, or magnetic stimuli, and on LCE containing photochromic mesogens and the mechanisms behind light actuation, better describing the current state of the art (to mention a few, see reviews from Schenning, Broer, Wiersma, Heggman, Ware, White, Ikeda, Priimagi, Yu, etc. group)

Reply 2.1: We thank the reviewer for reminding us to cite these important references and all relevant references mentioned by the reviewer have been cited in the Introduction of the Revised manuscript.

Comment 2.2:

Tuning T_i of polydomain xLCE-BP by annealing:

Neither in the text nor in Figure 1 is reported the temperature used for annealing below T_i (110°C?), while the initial T_i of the sample is mentioned within the text only at the end of figure 1d description (reporting the data at the beginning of the description would help the reader understanding the data). The same problem is present for figure 1e: annealing temperature? Since the annealing temperature is fundamental to modulate the T_i of the material, it should be clear both within the text and the figures the temperature used and reached (in terms of T_i) by the sample.

Reply 2.2: We are very sorry that we did not specify clearly the related annealing temperatures, the initial T_i and the final T_i for all figures in our old manuscript. Following the reviewer's advice, we have marked all the "initial T_i and final T_i " with detailed values in the Revised manuscript, and the corresponding annealing temperatures were also been added to all figures as well as the text in the Revised manuscript or the Revised Supplementary Information. Besides, we added the following to make it clear that the main material used in this work is polydomain xLCE-BP with a T_{i0} of 114°C:

"The unannealed polydomain xLCE-BP has a glass transition temperature (T_g) of 58°C and a T_i of 114°C on heating. All the following experiments used this material as the model original material before annealing unless otherwise noted. T_i of the original material before annealing was labeled as

T_{i0} and the annealing temperature is labeled as T_a in this work. Unless otherwise noted, the T_{i0} of the xLCE-BP used is 114°C.

Comment 2.3:

Why such a great difference in annealing below and above the T_i? Below the T_i, a quite close T is chosen (110 °C is quite close to 114 °C) while for annealing above a T_i much higher difference in temperature is used (180 °C is 66 °C higher than T_i). Which is the scientific explanation for this difference?

Reply 2.3: When we choose the annealing temperature, we consider two factors. One is the T_i. Below T_i, the material is in the LC phase; above T_i, it is in the isotropic phase. Annealing done below T_i allows the LC phases to grow. Annealing at a temperature higher than T_i can not help the perfection of the LC phase. **The other one is the exchange reaction.** The exchange reaction is fast at high temperatures while it is slow at low temperatures. To get a fast exchange, we use temperatures as high as possible so as to save the time which is necessary to fix the annealed structure. When annealing above T_i, 180°C is chosen as annealing at higher temperatures for a long time may lead to oxidation. When annealing below T_i, 110°C is the highest temperature that we may choose to ensure fast exchange reactions, otherwise, the annealed structure is hard to be fixed by the annealing. Because of the difference in the phase state, the fixed network topology of the material after annealing at different temperatures is very different, which leads to a great difference in the final T_i.

Thanks to the reviewer's comments, to make it clear to the readers, we added the following in the Revised Manuscript after the description on the change of T_i by annealing at 110°C and 180°C in the **"Tuning T_i of polydomain xLCE-BP by annealing"** section:

"When we choose the annealing temperature, we consider two factors. One is the T_i. Below T_i, the material is in the LC phase; above T_i, it is in the isotropic phase. According to previous reports of LC materials, annealing below T_i allows the LC phases to grow. The other one is the exchange reaction. The exchange reaction is fast at high temperatures while it is slow at low temperatures. To get a fast exchange rate, we use temperatures as high as possible. When annealing above T_i, 180°C is chosen as annealing at higher temperatures for a long time may lead to oxidation. When annealing below T_i, 110°C is the highest temperature that we choose to ensure fast exchange reactions below T_i. Because of the difference in the phase state, the fixed network topology of the material after annealing at different temperatures is very different, which leads to a great difference in the final T_i."

Comment 2.4:

Figure S1:

it would be beneficial to also have the structure or a representation of the final products

Reply 2.4: Following the reviewer's advice, the illustration on the structures of the final products have been added as FigureR1 (Figure S11 in the Revised Supplementary Information) as follows:

Figure R1. Illustration on the structural change of xLCE-BP after annealing

Comment 2.5:

Description of Figures S6c and d in the text are exchanged (Figure S6c reported data from Vitrimer-PU, not Vitrimer-BA)

Reply 2.5: We thank the reviewer for reminding us of the exchange for the description of Figures S6c and d in the old text. We have made the corresponding change.

Comment 2.6:

DSC traces reported in Figure S6b cannot be used to determine the T_g : no picks are evident from the reported traces, a DMA analysis will be more correct in this case.

Reply 2.6: Thanks to the reviewer's comment, we realized that we made a big mistake on the measuring temperatures range. We redid this experiment which showed a clear T_g and replaced the old figure with the new Figure R2 as the following, and the Figure R2 was added as the Figure S6 in the Revised Supplementary Information:

Figure R2. The DSC heating traces were used to determine T_g for Vitrimers-BA(a), Vitrimers-PU(b), and, PDMS elastomer(c) after annealing at 110°C for different time.

Comment 2.7: Stability and the reversibility of the new T_i . Figure 2b: what happens after 12 h? Longer experiments were done? Even in this section reporting in the text and the Figure explicating the temperatures mentioned would help the reader.

Reply 2.7: We repeated this experiment for longer time (Figure 2e in the“Revised manuscript”). The sample with a T_i of 77°C (obtained by annealing at 180°C for 4 days) decreased to 75°C after 12 hours of repeated DSC cycle tests with a scanning rate of $5^\circ\text{C}/\text{min}$. After continuing the cycle for another 61 hours, T_i dropped by only 3 degrees Celsius(from 75°C to 72°C). We have added the following:

“As shown in Figure 2f, the sample with a T_{i0} of 77°C (obtained by annealing xLCE-BP with T_{i0} of 114°C at 180°C for 4 days) decreased to 75°C after 12 hours of repeated DSC cycle tests from -10°C to 180°C with a scanning rate of $5^\circ\text{C}/\text{min}$. After continuing the test for 61 hours, T_i dropped by only 3 degrees (from 75°C to 72°C).”

Comment 2.8:

The generality of the method

This section does not explicitly report any T. From the supporting I gain the following:

xLCE-DHMS $T_i=81/84^\circ\text{C}$ (heating/cooling)

xLCE-PU $T_i=97/102^\circ\text{C}$ (heating/cooling)

xLCE-RM257 $T_i=79/88^\circ\text{C}$ (heating/cooling)

but these temperatures are referred to polydomain samples, while the description is reported for xLCE-DHMS in the smectic phase and xLCE-PU and xLCE-RM257 in the nematic phase. The author should explicit the temperatures within the text and explain to a broader readership that these temperature does not change in this case.

Reply 2.8: We thank the reviewer for the kind suggestion. We have added the following:

“According to previous reports, xLCE-DHMS has a smectic phase while xLCE-PU and xLCE-RM257 have nematic phases. No matter whether smectic or nematic, the obtained materials are all polydomain LCEs. On heating, the value of T_{i0} for xLCE-DHMS, xLCE-PU, and xLCE-RM257 is 84°C, 102°C, and 88°C respectively, and on cooling, the value of T_{i0} for xLCE-DHMS, xLCE-PU, and xLCE-RM257 is 81°C, 97°C, and 79°C respectively.”

Comment 2.9:

From table S3 I understood that all annealing experiments for the 3 samples were performed at 80°C, considering 80°C a lower T (with respect to T_i) for xLCE-DHMS and xLCE-PU and a higher T (with respect to T_i) for xLCE-RM257.

Why a so close (at least for xLCE-DHMS and xLCE-RM257) T was chosen? Would not be more correct to select at least 75°C and 85°C as lower or higher temperatures to have some degrees in between the annealing T and the T_i ? Why experiments comparable to the previous ones, therefore also performed at 65°C, 110°C, and 180°C were not repeated?

Reply 2.9.1: We agree with the reviewer that it is better to select at least 75°C and 85°C as lower or higher temperatures. We used to consider using those two temperatures, but we finally gave up and used the current choice. The major conclusions for our work are 1) the exchange reaction can fix the annealed structure; 2) annealing above T_i increase the value of T_i while annealing below T_i lowers the value of T_i . The experiment here is to verify those conclusions. **On one hand, we need experiments very similar to those of xLCE-BP; on the other hand, we want to show the above conclusion is also correct in other annealing conditions.**

For xLCE-DHMS ($T_{i0}=81^\circ\text{C}$), **we choose 80°C, which is similar to the annealing xLCE-BP($T_{i0}=114^\circ\text{C}$) at 110°C.** This is to make the experiment **comparable to the annealing of xLCE-BP.** In those experiments, we tried to choose a temperature as high as possible so long as it is below T_i . The higher the temperature, the faster the exchange reaction. If the temperature is too low, the exchange reaction will be too slow to be effective.

For xLCE-PU, we choose 80°C, which is much lower than its T_i (102°C). The reason is that we want to **show annealing at a temperature much lower than T_i also works so long as the exchange reaction is still fast.** As all the other experiments on annealing below T_i are done slightly lower than their respective T_{i0} , we need an experiment different from that.

For xLCE-RM257, we choose 80°C which is slightly above its T_i . **As we have already had an experiment of xLCE-DHMS annealing at 180°C and 140°C for different times, which is comparable to the annealing xLCE-BP at high temperatures as shown in Figure S9 in the Revised Supplementary Information, we here use xLCE-RM257 to show that even the annealing temperature is slightly higher than T_i , annealing above T_i still gives lower final T_i .**

Moreover, there is another reason is that we choose 80°C, which is that we can put all the samples together in the same oven to guarantee the annealing condition is the same.

Thanks to the reviewer's question, we added the following to make the above clear to the readers:

After the description of the annealing on xLCE-DHMS and xLCE-PU, we added:

“For xLCE-DHMS ($T_{i0} = 81^\circ\text{C}$), the choice of annealing temperature is similar to the annealing xLCE-BP ($T_{i0} = 114^\circ\text{C}$) at 110°C , that is, T_a is lower than T_{i0} while T_a is very close to T_{i0} . For xLCE-PU, the annealing temperature is much lower than T_{i0} , but the conclusion that annealing at $T_a < T_{i0}$ increase T_i still stand.”

After the description of annealing xLCE-RM257, we added: *“This experiment proves that annealing at a temperature slightly above T_{i0} can also decrease T_i .”*

Why experiments comparable to the previous ones, therefore also performed at 65°C, 110°C, and 180°C were not repeated?

Reply 2.9.2: The main aim of our experiments with xLCE-DHMS, xLCE-PU, and xLCE-RM25 is to verify the validity of our annealing method on xLCEs for tunable but stable T_i . Also, we want to verify that when annealing above T_i , the new T_i is below the initial T_i , while annealing below T_i resulted in increasing in T_i . So, **when choosing the annealing temperature, we tried to choose a temperature higher than T_i and a temperature lower than T_i . For different materials, the T_i is different.** Therefore, we did not try to perform all the annealing experiments at the same temperature. **As all the experiments here have verified our conclusion, we only used some temperatures as examples instead of doing a very detailed investigation on all the temperatures.** In this work, we haven't been able to clarify the effect of the exchange rate on the annealing, nor the effect of the liquid crystal phase on the annealing. We want to **leave those detailed investigations to another paper as this paper is only to show annealing xLCE is an effective method.**

Thanks to the reviewer's comment, we added the following to make it clear to the readers:

“As we can see from the three LCEs annealing at 80°C , even though the same annealing temperature is used, the change of T_i varies as the molecular structure changes. More systematic work is necessary to further clarify the effect of exchange reaction, molecular structure and phase behavior on the tuning of T_i by annealing.”

Figure S7 reports different sections and scales for the 3 traces, it would be better to have all the

scales from 60°C to 180°C with the same scale.

Reply 2.9.3: Following the reviewer's advice, Figure S7 in the old manuscript was replaced by a new Figure S8 in the Revised Supplementary Information, and the scales for the 3 traces in Figure S8 in the Revised manuscript were all the same. However, the data can not be plotted from 60°C to 180°C, otherwise, the peaks will be too small to be distinguished. We are sorry that we here can not follow the kind advice from the reviewer.

Comment 2.10:

Preparation and tuning of the actuation temperature of monodomain xLCE-BP

Figure 3b is partially described before and after figure 3c. It would be better to describe the figure all at the same point (before figure 3c). Moreover, the pictures on the right of the boxes are not well defined, maybe a higher contrast will help see the differences within the LCE strips.

Reply 2.10: We have revised the text according to the reviewer's kind suggestions.

Comment 2.11: The authors wrote: "The splits of the T_i peak can be observed on heating and cooling. Compared to polydomain samples, such a split gives out more sub-peaks." Without explaining and discussing this phenomenon. Please discuss this point.

Reply 2.11: Honestly, we are not sure about the exact reason. We checked some papers and found that the alignment affected the annealing. So, thanks to the reviewer's kind advice, we added the following to explain:

1) When talking about the splits on the T_i for the first time, we added:

"On both heating and cooling curves, the T_i peak splits as the annealing time extends. Similar splits have been observed in other LC polymers when annealed near T_i ³⁸ Such splits were attributed to the polydispersity in the molecular mass. Even though the material here has a network structure, the molecular weight between cross-links has similar polydispersity. The presence of transesterification enables the network to act like linear polymers and the structure rearranged during annealing."

1) When discussing the T_i splits of monodomain, we added: *"Probably the alignment of the mesogens further promoted the polymorphism⁴⁴"*

Comment 2.12:

Furthermore, relating to the phrase: "even after 200 rapid heating-cooling cycles on a hot plate, the actuation strain of the sample with a T_i of 150°C hardly changed (Figure 3e)" what is considered "rapid"? times? Please better detail, explain and discuss this aspect!

Reply 2.12: Thanks to the reviewer's kind advice, we added: *"In the above 200 cycles, it takes about 7 minutes to complete a contraction-elongation cycle between 140°C and 160°C. The heating process takes about 2 minutes and the cooling process takes about 5 minutes. This contraction-elongation cycle is carried out for 200 times in total (one cycle is shown in video S1)"*

Comment 2.13: About Figure 3: temperatures and behavior at 0d should be made explicit in Figure 3c and 3b, respectively. Panel e is too small to be understandable and scales are missing.

Reply 2.13: Following the reviewer's advice, Figure 3 in the old manuscript was replaced by the new one. In the new Figure 3b (Figure 3c in the old manuscript), the initial T_i at 0d was added in the DSC traces, and the panel is enlarged to see the changes of all samples more clearly in the Figure 3f (replacing the Figure 3b in the old manuscript) in the Revised manuscript. In addition, the missing scales and the sample preparation methods were also added in the Figure 3f in Revised manuscript.

For the sample used in Figure 3f, at 0 days, it is polydomain xLCE-BP ($T_{i0}=114^\circ\text{C}$) which has no actuation.

To make it clear to the readers, we added:

"The obtained xLCE-BP is polydomain (no macroscopic orientation and no actuation)" after we described the synthesis of xLCE-BP.

When describing figure 3f, we added: *"The sample with T_i of 90°C (T_{i1}) and 112°C (T_{i2}) were obtained by annealing polydomain xLCE-BP ($T_{i0} = 114^\circ\text{C}$) after stretching at 180°C for 24h and 3h, respectively. The samples with T_i of 122°C (T_{i3}), 135°C (T_{i4}) and 152°C (T_{i5}) were obtained by annealing xLCE-BP ($T_{i0} = 114^\circ\text{C}$) at 110°C after stretching for 3 days, 10 days and 30 days, respectively. The five samples are placed together."*

Comment 2.14: Investigation of the structure change during annealing by XRD. Also in this case is very difficult to understand the annealing conditions of the data reported!

Reply 2.14: We are sorry that we did not make it clear enough. We have specified all the annealing conditions in the figures.

Comment 2.15: The author wrote: "Figure 4b shows the dependence of the lamellar spacing on annealing time at respective temperatures." Where are these temperatures indicated? I cannot find them neither in the text nor Figure 4b.

Reply 2.15: Thanks to the reviewer's kind reminder, we added the related temperatures as shown in Figure R3 (Figure S10 in the Revised Supplementary Information) and specified the annealing temperature and time.

Figure R3. The change of d-spacing of polydomain of xLCE-BP ($T_{i0} = 114^{\circ}\text{C}$) after annealing at 110°C from 0d to 50d.

Comment 2.16: Furthermore: “In the first 15 days, the spacing decreases as the annealing time is increased and reaches 28.15 \AA on the 15th day.” It looks to me that this is true after 5 and not 15 days. “It looks like that annealing lower than T_i helps to the formation of a relatively more compact layered structure, but loner time annealing may not be good.” Also in this case, annealing conditions? T_i ? Where these data are reported?

Reply 2.16: For xLCE-BP ($T_{i0}=114^{\circ}\text{C}$), when it was annealed at 110°C , the values of 2θ after annealing at 5d and 10d is the same. At 15d, it decreases again. The reviewer is right. It is more appropriate to say “after 5 ” instead of “ 15 days”. We adopted the reviewer’s opinion in the Revised Manuscript. All the detailed XRD data, annealing conditions and T_i are provided in table S6 and table S7 in the Revised Supplementary Information.

Comment 2.17: About the comparison of the experiments performed at 110°C , 180°C and 140°C why they do not report the corresponding data also at the same time?

Reply 2.17: Thanks to the reviewer's question, we also reported the corresponding data in Table S6 in the Revised Supporting Information.

Comment 2.18: Figure 4f only describes the data recorded after annealing at 180°C for 24h and not the data after annealing 3d at 110°C . Moreover, the description reported looks more appropriate to the sample annealed at 110°C for 3 days.

Reply 2.18: Figure 4f in the old manuscript has been reassembled in Figure 4e in the Revised manuscript. Figure 4e contains all one-dimensional XRD data of monodomains obtained by annealing polydomain xLCE ($T_{i0} = 114^{\circ}\text{C}$) above T_i (annealing at 180°C for 24h) and below T_i (annealing at 110°C from 0 to 50 days). Now the description is consistent with the figures.

Reviewer #2 (Remarks to the Author):

This is an exciting study demonstrating a simple and general strategy for tuning the isotropization temperature (T_i) for liquid crystalline elastomers (LCEs). T_i is a very important parameter for LCEs as it determines the actuation temperature and related properties for LCE functional applications. The traditional methods for changing T_i of a LCE is through modifying its chemical structure, which is tedious and not general. While physical methods like annealing can induce the change of T_i , one major issue is that the new T_i after annealing immediately goes back to the old one when heated above the T_i , which limits its practical applicability. As the authors pointed out in the introduction, there is no effective strategy to tune the actuation temperature without changing the chemical composition so far. In this study, the authors made a serendipitous discovery demonstrating that the dynamic covalent bonds and annealing together make it possible to tune T_i after the material is fully covalently crosslinked. The key to achieve this tunability of T_i is through the use of dynamic covalent crosslinks to the LCEs. In nutshell, the tunability was achieved by taking advantage of two synergistic thermal effects: 1) thermal annealing to tune actuation temperature, and 2) dynamic covalent bond exchange during annealing and subsequent fixing upon removal of the heat. In contrast to normal LCEs that lose new T_i after heating up the annealed samples, in the new systems reported here different T_i 's achieved by annealing can be preserved by reversible reaction of dynamic covalent bonds in covalently adaptable LC networks including LC vitrimers. The authors further demonstrated the generality of this approach by showing similar behavior in several LCE systems using different dynamic covalent bonds including transesterification, urethane exchange, etc. The authors did an excellent job in showing the tunability of T_i over a broad range by simple annealing. They also showed convincingly that the annealed samples show good stability for the new actuation temperature over many heating/cooling cycles. The structural change during annealing of the LCE samples were carefully monitored by XRD. Lastly, the authors creatively showed patterning on LCE film by precisely controlling annealing using digital patterning and electrothermal films. Overall, this work demonstrates an exciting new and general methods for tuning the actuation temperature of LCE materials, which makes an important contribution to the dynamic polymer materials area. This reviewer would enthusiastically support the acceptance of this work by Nature Communication after addressing the following minor issues.

Reply: We thank the reviewer very much for the positive comment on our work. We also appreciate the concerns that the reviewer raised, which helped us a lot to improve the work.

Comment 1:

In the discussion, the authors should make it more clear how physical annealing changes the actuation temperatures for LCEs. The authors cited some references in the manuscript for this effect but it would help readers a lot by briefly discussing how annealing at different temperatures changes the structure differently and accordingly the actuation temperature of LCEs.

Reply 1: We thank the reviewer for reminding us about this discussion. It will be very helpful for our readers. We added the following to give our readers a general understanding of annealing.

In the introduction part, we added: “Annealing is a well-established heat treatment process that can change the microstructure of the material by leaving the material at a certain temperature for some time.²⁷ Even though it is possible to change T_i by annealing after synthesis due to the structure change during annealing,” and “Here, unlike all the previous annealing of LCEs, the new structure obtained by annealing can be fixed thanks to the exchange reaction that occurred during the annealing.”

When discussing the annealing temperature, we added “Because of the difference in the phase state, the fixed network topology of the material after annealing at different temperatures is very different, which leads to a great difference in the final T_i ”

After the X-ray analysis, we added: “Therefore, annealing changes the structure of the network. Annealing at $T_a < T_{i0}$ helps to the formation of a relatively more compact layered structure while annealing at $T_a > T_{i0}$ impairs the regularity of lamellar stacking as illustrated in Figure R1 (Figure S11 in the Revised Supplementary Information).”

Figure R1. Illustration on the structural change of xLCE-BP after annealing

Comment 2:

Some figures/data are a little hard to follow which need some revision/improvement. For example, in Fig. 3b, it takes multiple reading to figure out the experiments. Perhaps label T_1 , T_2 , T_3 ,... next to each sample.

Reply 2: Following the reviewer’s advice, we have improved the image formatting and data clarity. For example, in the new Figure 3h in the Revised manuscript, the monodomain samples with different actuation temperatures were labeled as T_{i1} , T_{i2} , T_{i3} , T_{i4} , and T_{i5} , respectively.

Comment 3:

- Some writing expressions need to be revised. For example, pp 3, line 6: “this highly bright promise” “this promise”. Pp18, line 8: “In Figure 3b shows” “Figure 3b shows”.

Reply 3: Following the reviewer's advice, we have tried our best to carefully revise the whole manuscript.

Reviewer #3 (Remarks to the Author):

Comment

The manuscript identifies a new methodology for tuning the nematic to isotropic phase transition temperature in isotropic-genesis and mechanically aligned LCE materials containing covalent adaptable networks (xLCEs). The methodology enables the transition temperature to be tuned in a relatively wide range of temperatures. The authors provide evidence of their deterministic ability to tune the temperature to their desired values across a range of xLCE materials and confirm their results using a combination of experimental methods, the main of which are DSC and XRD. To the best of this reviewer's knowledge, the results presented in this work are original, the used chemical compositions and experimental methodologies are scientifically sound and the obtained results are correct, new, interesting, and complement the contemporary literature in the field. I, therefore, recommend the publication of this manuscript in Nature Communications with the following comments:

Reply: We thank the reviewer for the positive comment on our work. We also appreciate the following concerns that the reviewer raised, which is also very helpful to improve the manuscript. Following the reviewer's advice, we have made corresponding changes in the text.

Comment 1

1. Significant changes to the shape of the Ti peak (involving phenomena such as broadening and splitting), as well as significant changes to the Tg (along one direction of the annealing process) were observed throughout the first section of the manuscript. The authors were further able to demonstrate that their methodology does not apply to non-LC and non-CAN materials. However, although the reversibility of the process is announced, the changes in peak shape indicate that the material does change in certain ways (e.g., Figure 2f). It is my opinion that the manuscript would benefit from an additional discussion of this issue.)

Reply 1: We thank the reviewer for the kind suggestion. Following this suggestion, we added:

In the introduction part, we added: "Annealing is a well-established heat treatment process that can change the microstructure of the material by leaving the material at a certain temperature for some time.²⁷ Even though it is possible to change T_i by annealing after synthesis due to the structure change during annealing," and "Here, unlike all the previous annealing of LCEs, the new structure obtained by annealing can be fixed thanks to the exchange reaction that occurred during the annealing."

When discussing the annealing temperature, we added "Because of the difference in the phase state, the fixed network topology of the material after annealing at different temperatures is very

different, which leads to a great difference in the final T_i ”

After the X-ray analysis, we added: “Therefore, annealing changes the structure of the network. Annealing at $T_a < T_{i0}$ helps to the formation of a relatively more compact layered structure while annealing at $T_a > T_{i0}$ impairs the regularity of lamellar stacking as illustrated in Figure R1 (Figure S11 in the Revised Supplementary Information).”

Figure R1. Illustration on the structural change of xLCE-BP after annealing

Comment 2

2. The XRD results do provide the necessary evidence for the statements that have been made. However, in figures 4e and 4f, the specific choice of annealing temperatures (110C and 180C) is somewhat unclear. A short explanation qualifying these choices of evaluation temperatures in terms of their effects on T_i and their comparability would do well to strengthen the authors' point.

Reply 2: Thanks to the reviewer's comment, we added the following:

“When we choose the annealing temperature, we consider two factors. One is the T_i . Below T_i , the material is in the LC phase; above T_i , it is in the isotropic phase. According to previous reports of LC materials, annealing below T_i allows the LC phases to grow. The other one is the exchange reaction. The exchange reaction is fast at high temperatures while it is slow at low temperatures. To get a fast exchange rate, we use temperatures as high as possible. When annealing above T_i , 180°C is chosen as annealing at higher temperatures for a long time may lead to oxidation. When annealing below T_i , 110°C is the highest temperature that we choose to ensure fast exchange reactions below T_i . Because of the difference in the phase state, the fixed network topology of the material after annealing at different temperatures is very different, which leads to a great difference in the final T_i .”

Comment 3

3. The main experimental details of the XRD method should be stated directly in the main text and not only in the supplementary. Figure 4 would benefit from adding a scalebar for the reciprocal space

or some notation for the obtained reflections.

Reply 3: Following the reviewer's advice, we added the experimental details of the XRD method in the Revised manuscript. The obtained reflections were also noted in the text and the detailed data has been summarized in Table S6 and S7 in the Revised Supplementary Information.

Comment 4

5. The authors are encouraged to perform an additional language proof to bring the writing up to the journal's standards.

Reply 4: We are very sorry for our bad English. We have asked a colleague to help us with this.

REVIEWERS' COMMENTS

Reviewer #2 (Remarks to the Author):

The authors adequately addressed the minor concerns from this reviewer, so I recommend acceptance of this manuscript by Nat Comm.

Reviewer #3 (Remarks to the Author):

The authors have improved their manuscript significantly and addressed all the raised issues. I therefore recommend that the manuscript be accepted for publication in the journal.